# Extended Reality Technologies in Nutrition Education and Behavior: Comprehensive Scoping Review and Future Directions

**DOI:** 10.3390/nu12092899

**Published:** 2020-09-22

**Authors:** Jared T. McGuirt, Natalie K. Cooke, Marissa Burgermaster, Basheerah Enahora, Grace Huebner, Yu Meng, Gina Tripicchio, Omari Dyson, Virginia C. Stage, Siew Sun Wong

**Affiliations:** 1Department of Nutrition, School of Health and Human Sciences, University of North Carolina at Greensboro, Greensboro, NC 27412, USA; brenahor@uncg.edu (B.E.); gracehueb@gmail.com (G.H.); 2Department of Food, Bioprocessing and Nutrition Sciences, North Carolina State University, Raleigh, NC 27695, USA; nkcooke@ncsu.edu; 3Departments of Nutritional Sciences and Population Health, The University of Texas at Austin, Austin, TX 78712, USA; marissa.burgermaster@austin.utexas.edu; 4Imperial County Cooperative Extension, University of California Agriculture and Natural Resources, Holtville, CA 92250, USA; ucmeng@ucanr.edu; 5Center for Obesity Research and Education, College of Public Health, Temple University, Philadelphia, PA 19140, USA; gina.tripicchio@temple.edu; 6Department of Peace and Conflict Studies, School of Health and Human Sciences, University of North Carolina at Greensboro, Greensboro, NC 27412, USA; oldyson@uncg.edu; 7Department of Nutrition Science, College of Allied Health Sciences, East Carolina University, Greenville, NC 27858, USA; carrawaystagev@ecu.edu; 8College of Public Health and Human Sciences, Oregon State University, Corvallis, OR 97331, USA; siewsun.wong@oregonstate.edu

**Keywords:** extended reality, virtual reality, augmented realty, mixed reality, nutrition education, nutrition behavior, digital technology, scoping review

## Abstract

The use of Extended Reality (XR) (i.e. Virtual and Augmented Reality) for nutrition education and behavior change has not been comprehensively reviewed. This paper presents findings from a scoping review of current published research. Articles (*n* = 92) were extracted from PubMed and Scopus using a structured search strategy and selection approach. Pertinent study information was extracted using a standardized data collection form. Each article was independently reviewed and coded by two members of the research team, who then met to resolve any coding discrepancies. There is an increasing trend in publication in this area, mostly regarding Virtual Reality. Most studies used developmental testing in a lab setting, employed descriptive or observational methods, and focused on momentary behavior change like food selection rather than education. The growth and diversity of XR studies suggest the potential of this approach. There is a need and opportunity for more XR technology focused on children and other foundational theoretical determinants of behavior change to be addressed within nutrition education. Our findings suggest that XR technology is a burgeoning approach in the field of nutrition, but important gaps remain, including inadequate methodological rigor, community application, and assessment of the impact on dietary behaviors.

## 1. Introduction

Many people face significant barriers to participating in traditional nutrition education programming, including time, transportation, and family responsibilities, which often leads to isolation and disconnectedness, particularly in rural or low-income areas [1,2]. Most programs are only available on a limited basis (i.e., once per week or less) and provide a potentially insufficient intervention dose, especially with limited attendance. These factors contribute to a low motivation to participate, though this may be overcome through positive program experiences [3]. The COVID-19 pandemic has also demonstrated the need for virtual approaches to nutrition education and behavior change programs that can be accessed from remote settings [4,5]. 

The standard delivery of nutrition education in the United States (e.g., in-person, face-to-face) still significantly lags in the integration of technological advances, which may limit its success in meeting the diverse needs and interests of participants, especially those from low-income communities [6]. There has been a call to increase online nutrition education opportunities for limited-resource learners [6]. One possible solution is innovative computer software technologies like Extended Reality (XR), which includes the subcategories of Augmented (AR), Virtual (VR), and Mixed Reality (MR) [7]. XR technology offers the flexibility needed in a platform to address the wide-ranging needs of a nutritional education and obesity prevention program while offering technological aspects of the programming that could be enticing to many families, children, and youth. These types of technologies are available at different price points with varying degrees of immersion and physical-world applications. VR is an interactive computer-generated experience that takes place within a simulated environment [8]. VR can be accessed with a computer program (e.g., with 3D models or avatars), simple VR viewers connected to mobile phones, or with a more dedicated head-mounted display or helmet-mounted display. The VR environment incorporates mainly auditory and visual information [8]. This immersive environment can be like the real world or it can be fantastical, creating an experience that is not possible in ordinary physical reality [8]. AR is like VR but also superimposes a computer-generated image on a user’s view of the real world, thus providing a composite view [9]. AR can be accessed using mobile phone applications, AR smart eyeglasses, or dedicated headsets. MR is like AR, but the digital content is spatially aware and responsive so that virtual objects become part of the real world [10]. MR is accessed using a dedicated MR headset. More easily accessed and affordable devices like computer experiences and mobile phone applications increase accessibility but may decrease immersion and user sense of presence compared to more expensive headset devices. Greater user presence may enhance applied effectiveness [11]. A meta-analysis of the effect of immersive technology on user presence found that technological immersion has a medium-sized effect on presence, and that individual immersive features such as user-tracking, stereoscopic visuals, and wider fields of view of visual display are particularly impactful features on the level of presence [11]. 

These technologies can revolutionize nutrition education and behavior change approaches due to the ability to dynamically increase access to knowledge and experiences that may not be accessible for many. Other potential attributes include motivational reinforcement, personalized teaching approaches, and social networking opportunities. As seen during the COVID-19 pandemic, access to traditional face-to-face forms of nutrition education and behavior change programming can be severely impacted by limiting environmental and contextual factors. Now is the time to understand how XR technologies can help maintain access to nutrition education and behavior change programming when face-to-face programs are not accessible. At this point, it is unclear how these technologies have been used for nutrition education and behavior change and what gaps remain in the development and implementation of XR programming. 

Thus, there is a need for an examination of the literature around the use of XR technologies in nutrition education and behavior change to identify gaps and needs for future work. A previous paper [12], which summarized the background information and discussions for a joint July 2010 National Institutes of Health—US Department of Defense workshop, discussed the research needs and opportunities of VR technologies for research and education in obesity and diabetes, but the report focused solely on VR for obesity and diabetes. Since the publication of that report, various iterations of XR, including AR and MR, have emerged, and the potential of this technology has been more fully realized. 

The purpose of this paper was to conduct a scoping review of all published research related to the use of XR technologies for nutrition education and behavior change to determine how these technologies have been used in this field and what the future needs are for this field of research. 

## 2. Materials and Methods 

### 2.1. Review Process

The methods for this comprehensive scoping review were guided by the Preferred Reporting Items for Systematic Reviews and Meta-Analyses (PRISMA) [13]. The goal of this approach is to minimize bias through systematically identifying, selecting, and critiquing relevant research. For this paper, we used a scoping review process, which is a rigorous and transparent method that seeks to provide an overview of a potentially large and diverse body of literature pertaining to a broad topic, rather than a systematic review which attempts to collate empirical evidence from a relatively smaller number of studies pertaining to focused research questions [14]. This approach is of particular use when the topic has not yet been extensively reviewed or is of a complex or heterogeneous nature, which is the case in this topic [15]. 

### 2.2. Search Strategy

We conducted an extensive search of online journals and research databases, focusing on PubMed and Scopus results. Our search of the literature included two domains: XR approaches and dietary behaviors. We developed a search strategy with the guidance of an experienced academic research health sciences librarian, ultimately using Boolean operators and searched within subject heading, title, and abstract field. The search terms were developed after expert review and consensus and included: augmented reality or virtual reality or mixed reality or avatar AND diet or nutrition or buffet or restaurant or grocery AND NOT farm. We searched in the English language, with a time filter of the past decade (1/1/2009 and later); articles had to be referenced in the research databases we selected and meet the inclusion criteria. To reduce potential selection bias, we applied no restrictions on article type or clinical population.

### 2.3. Selection of Articles

We included all research design types, including randomized controlled trials, quasi-experimental studies, observational studies, and product development/user testing. We focused on studies that used XR for nutrition education and nutrition behavior change as both the primary focus or as the means to an end (the technology was used to achieve a condition with the experience or impact of XR not being the focus). We excluded research that was a review of previous research that did not have research subjects and research that did not pertain to nutrition education or behaviors.

### 2.4. Data Extraction

The research team developed a detailed process for data extraction and then aggregated a list of all studies that met the eligibility criteria. Members of the team then drafted an initial version of a standardized, detailed data collection form in REDCap (Research Electronic Data Capture) [16]. REDCap is a secure, web-based software platform designed to support data capture for research studies [16]. The study team reviewed a sample of three articles with the initial version of the data collection form and then met to discuss necessary changes to the form. Using feedback, the research team finalized the data extraction form before implementation. One researcher made data extraction assignments to ensure that each article was independently reviewed and coded by two members of the research team, who then met to resolve any coding discrepancies. Coding teams determined discrepancies by going through each of the coders’ initial submissions for each article and highlighting any discrepancies in answers, coming to a consensus through coder meeting. When issues arose or could not be resolved, the larger research team discussed them and came to a consensus. 

The following information was extracted from each of the reviewed studies: reviewer name, review date, article title, article first author, journal name, date published, article type (full text or abstract), mode of technology used (augmented reality, mixed reality, and/or virtual reality), research focus (nutrition education, behavior change, consumer preferences, product testing, narrative review, other), XR purpose (XR was the focus, XR was only a means to an end (delivery method only)), location of project (international, United States, both international and United States, unclear), setting of project (community (research conducted in non-lab, non-clinic, and non-retail settings at community sites), lab, retail, clinic, other), sample size (number of participants), age of participants, race of participants, whether health disparities were mentioned, whether participant income was measured (yes/no), focused on low-income population (yes/no), longitudinal or non-longitudinal, duration of the study (for either longitudinal or non-longitudinal), study design (randomized controlled trial, quasi-experimental, other controlled trial, single group pre-post, descriptive or observational, qualitative, other, not reported, validation), intervention study (yes/no), intervention details (open-ended), type of device used (open-ended), main outcome measures (diet, purchasing, selection of option (non-purchasing), knowledge, self-Efficacy, experiences, behaviors, engagement (which involves interaction), other), suggested mechanism for the behavior or knowledge change (open-ended), whether perceived user presence in the virtual environment was measured (yes/no), main results (open-ended), study outcomes (AR/VR improved outcomes compared to real life; No difference between AR/VR and real life; AR/VR had negative outcomes compared to real life; Does not apply), measured user fatigue (yes/no), and study conclusions (open-ended).

### 2.5. Data Analysis 

Researchers generated descriptive statistics for data collection form variables using SPSS and calculated frequencies, percentages, and means and standard deviations based on variable type. We examined the relationship between (1) XR technology type (AR, VR, MR) and study objective (nutrition education, behavior change, consumer preference, product testing, concept paper, other); (2) technology type and study outcome (diet, purchasing, selection, knowledge, self-efficacy, experiences, behaviors, engagement, other); and (3) technology type and impact on nutrition education or behavior (improved outcomes compared to real life, no difference, and negative outcome compared to real life) using cross tabulation frequencies. 

### 2.6. Data Availability 

For a listing of reviewed manuscripts and data not shown in the Results section, please see the Appendix A. 

## 3. Results

### 3.1. Review Characteristics

The process for identifying and selecting papers for review is found in Figure 1. The initial search of the research databases identified 116 articles. After review, 15 narrative reviews and four protocol papers were excluded. Three duplicate studies were also removed. Finally, two studies in which VR was used to measure a non-nutrition-related outcome (embryonic development) were removed. The research team then extracted and coded the resulting list of 92 unique studies.

### 3.2. General Characteristics of Articles

A summary of general characteristics of articles can be found in Figure 2. Most of the articles focused on VR (*n* = 50), followed by AR (*n* = 36) and MR (*n* = 6). Few studies reported utilizing an avatar (13/92; 14.1%). Most studies had an ER experience as the focus (*n* = 71), but a fair number of studies only used XR as a means to an end (*n* = 21), primarily to create a food-related environment or situation for a non-nutrition behavior research question. There was a general increasing trend of articles published on this topic between 2005 and 2018 (2019 was excluded due to articles being extracted halfway through the year), with the most articles published in 2017 (*n* = 18) (see Figure 3).

The average study size of articles reporting sample sizes (*n* = 9) was 79.1 (SD = 106.2) participants. Few studies were focused on children and teens (*n* = 7) or older adults (*n* = 5). For most studies (*n* = 71), the racial composition of participants was unreported or unclear. Few studies reported income information (*n* = 7), and only 3 studies reported focusing on low-income populations. 

Most studies were conducted in an international setting (*n* = 62) rather than the US (*n* = 20). Lab settings were the most common location of research (*n* = 53) because many studies were in the development stage. Community settings (*n* = 18) and Clinics (*n* = 13) were the next most common setting, mostly due to the ability to recruit participants in this setting, with retail settings being the least-reported setting (*n* = 4). 

The most common study design was descriptive or observational (*n* = 31), followed by randomized control trials (*n* = 19), quasi-experimental (*n* = 15), and qualitative studies (*n* = 8). The most common devices used for the XR experience included computers (*n* = 34), mobile apps (*n* = 25), head-mounted devices (*n* = 17), glasses (*n* = 4), smart utensils (*n* = 3), smart tables (*n* = 2), gaming devices (*n* = 2), and a few studies were unclear (*n* = 7). Many studies (*n* = 25) used multiple devices to create their user experience. 

### 3.3. Study Design, Objectives, and Outcomes

The most common research focus was behavior change (*n* = 50), followed by product testing (*n* = 4), and nutrition education (*n* = 22). Other areas of research focus included topics like cravings, memory, and eating disorder treatment. There were more studies with multiple identified focuses of research (*n* = 54) rather than a single focus (*n* = 38). 

Behavior change studies focused on food purchases/selection (*n* = 9) in virtual retail settings (e.g. supermarkets, buffets) (*n* = 7) and non-retail (*n* = 2), food/beverage consumption (*n* = 8; 2 focused on binge eating), changing cravings (*n* = 6), weight loss (*n* = 4), and program participation (*n* = 2), among others. Knowledge improvement studies primarily focused on food size/portion/serving estimation (*n* = 4), with only a few studies focused on nutrition guidelines (*n* = 1), diet monitoring (*n* = 1), diabetes self-management (*n* = 1), healthy food options (*n* = 1), or goal setting (*n* = 1). 

In examining study objectives by technology type (see Figure 4), the most common AR study objective was product testing (*n* = 22), followed by behavior change (*n* = 12) and consumer preference (*n* = 10). The most common VR technology study objectives were behavior change (*n* = 35), product testing (*n* = 19), and nutrition education (*n* = 12). The most common MR study objective was product testing (*n* = 3). 

The most common outcomes were behaviors (*n* = 57), experiences (*n* = 24), and knowledge (*n* = 19). We found the same order of prevalence of outcomes across AR and VR (see Figure 5). The most common AR outcomes were behaviors (*n* = 15), experiences (*n* = 7), and knowledge (*n* = 7). The most common VR outcomes were behaviors (*n* = 39), experiences (*n* = 15), and knowledge (*n* = 12). Only two studies reported information on fatigue, focusing on visual fatigue. Perceived user presence in the virtual environment was mentioned by relatively few studies (*n* = 13).

### 3.4. Impacts on Nutrition Education or Behavior

For the studies where impact could be assessed (the majority of studies did not apply), most studies improved outcomes compared to real life (*n* = 37), some showed no difference (*n* = 10), and none were found to have a negative outcome compared to real life. The distribution of outcomes across the technologies were largely the same. AR had 18 studies that improved outcomes compared to real life and two that made no difference compared to real life. VR had 20 studies that improved outcomes compared to real life and eight that made no difference compared to real life. MR had two studies that improved outcomes compared to real life, and four studies where impact could not be assessed. 

## 4. Discussion

Our study found an increasing focus in the published literature in the use of XR over the past 10 years. Most of the research has focused on VR as the mode of delivery, used an adult population, was located in international locations, and was conducted in lab settings. Most research used an observational or descriptive study design, focused on behavior change across a wide variety of topics (food purchases and selection, food/beverage consumption (including binge eating), and changing cravings), and found that AR/VR improved outcomes compared to everyday life. The increased focus of product development and research in this area suggests that this is an emerging and promising approach for nutrition education and behavior change. 

The predominant focus of studies was on short-term behavioral change in product selection, particularly for the retail environment. Studies also focused on virtual retail settings (e.g., supermarkets and buffets) rather than physical retail settings; therefore, more research around the integration of XR technologies in the physical retail setting is needed. This focus on the intersection of technology and the food environment suggests that this may be a practical focus area with the potential to change the healthiness of point of purchase or product selection by consumers. This approach builds off of the concepts of Behavioral Economics, including choice architecture (influencing choice by organizing the context in which people make decisions) and behavioral nudges (any aspect of the choice architecture that alters people’s behavior in a predictable way without forbidding any options or significantly changing their economic incentives), to try and influence health behaviors with environmental cues, or in this case, an artificially enhanced environment [17]. Most knowledge-based studies focused on food size/portion/serving estimation, with few studies focused on practical and foundational topics like nutrition guidelines, healthy food management, and diet management. While improving portion size estimation is an important skill, and the focus on this topic could be due to the intrinsic spatial nature of food/portion size estimation, there is a need and an opportunity to focus on other theoretical determinants of behavior change addressed within nutrition education (e.g., attitudes, social norms, self-efficacy, etc.) to promote healthier dietary habits.

Most studies focused on adult populations, rather than older or younger populations. More product development and research focused on non-adult populations, including children, teens, and older adults, is needed. Children and teenagers may be particularly interested in the use of VR and AR technology as a form of entertainment and may be more receptive to nutrition education and behavior change approaches which incorporate this technology [18,19]. Previous research has found that teenagers were interested in a VR avatar approach to weight loss because it would fit with their lifestyle and help support healthy behaviors [19]. The use of VR and AR may also be a desirable and effective approach for nutrition education and behavior change for older adults who have mobility limitations or limited access to in-person nutrition education programs [20]. There has been an increase in the availability and use of VR in senior living and memory care facilities due to the potential of this approach. Lin et al. in 2018 conducted a field study to understand how use of a VR system may contribute to older adults’ emotional and social well-being. They found that a VR application provided more benefits compared to the control condition and that participants that used the VR system reported being less socially isolated, being less likely to show signs of depression, experienced positive affects more frequently, and felt better about their overall well-being [21]. 

Most of the research we reviewed was observational or descriptive in nature. Many studies focused on conceptual ideas with promising potential but limited evidence for impact. There is a need for more studies in this area to utilize a more rigorous methodological approach in order to establish causality of effect more clearly. There is also a need to focus more on short- and long-term impacts on dietary behaviors and cardiometabolic risk, rather than the current focus on shorter term changes in knowledge, preferences, and skills. Additionally, more translational research is needed, particularly in the community setting where large scale implementation and impact could occur. While the use of lab settings is appropriate for product development, as these approaches become easier to develop, more community-based translational studies are needed to achieve an impact on population nutrition. This speaks to the need for more interdisciplinary teams that include community-engaged nutrition education and behavior change researchers, medical care teams, computer scientists, and other relevant disciplines. 

We found that very few studies reported sociodemographic characteristics in their samples. Most studies did not report detailed information on race, gender, or income, and very few studies reported health disparities or low-income populations being a focus of their research. It is important to understand cultural acceptance preferences for certain technology modalities and programmatic approaches. It is also important to understand income-related access to these technologies, including access to adequate internet, hardware, and software, and how these factors might influence utilization and impact. 

Our findings suggest similar impacts of the different technology modalities (AR, VR, MR) on improving experiences beyond normal life, but more research is needed that directly compares and contrasts the effectiveness of each technology modality for nutrition education and behavior change outcomes (e.g., determinants of behavior change, dietary patterns, weight management). The uniqueness of the potential application of AR, VR, and MR means that in many use cases and across different settings, one technology modality may be more appropriate and practical than another modality. Appropriateness is based on the potential impact the developer thinks the program will have on the specific outcome they are trying to achieve, whether it be behavior change or knowledge transformation. For instance, AR may be more appropriate than VR for product selection in a retail setting, but VR could be effective at influencing future product selection during practice for behaviors in the retail environment. Future work should try and clarify the advantages and disadvantages of each modality, across different settings, and with different areas of focus to help inform future programming efforts.

The practical use of this technology for nutrition education and behavior is important in order to achieve its wide-scale use in community settings. We found that very few articles reported usability factors like fatigue (*n* = 2) from performing necessary repetitive tasks, headset discomfort, headache, nausea, and eye strain from screen viewing. Fatigue could influence the duration of use and enjoyment of using device-based technologies like XR [22,23,24]. These issues could vary based on XR modality and tasks required for the experience. Considerations of fatigue should be made for all end-users including consumers and education program participants of varying age groups, as well as counselors, educators, and clinicians. A recent study among a sample of children using AR found that fatigue was highest under the gesture condition, and lowest under the controller condition, and that this led to controller conditions being reported as the most usable by participants [25]. A study of VR devices found that despite advances in technology, head-mounted displays and helmet-mounted display systems create visual discomfort after extended usage [26]. To determine if periods of rest could help visual fatigue during head-mounted display VR experiences, Guo *et al*. found that short rest during continuous use of the head-mounted display induced more severe symptoms of subjective visual discomfort, but reduced objective visual fatigue [27]. Future research should continue to examine this issue of fatigue and ways that technology can reduce it. This will be particularly important for the wide-scale uptake of XR nutrition education and behavior approaches in community settings. 

This study has limitations. One common limitation of scoping reviews of the literature is publication bias, as the papers which we reviewed may have been published due to their having certain outcomes, which could lead to an unbalanced reporting of findings and a bias in favor of positive results [28]. There also is the possibility that our review missed some relevant studies due to database selection, exclusion of gray literature, and exclusion of studies not published in English. The strengths of the study include the comprehensive review of the literature, the use of a structured and detailed data collection form, and the use of independent double coders. To reduce bias, we used a comprehensive and structured search and identified and included all papers that met our inclusion and exclusion criteria.

## 5. Conclusions

Our findings suggest that the integration of XR technology in the field of nutrition education and behavior change is a burgeoning approach that may have the potential to be successfully used to influence diet-related behaviors, though the efficacy and effectiveness of the impact on diet-related behaviors remains unclear. Future research should aim to build on the existing research by using more rigorous methodology, engaging more diverse populations including low-income individuals, children, and older adults, and should include a more focused assessment of the impact on dietary behaviors. Nutrition education and behavior change programs, including higher education and federal nutrition education programs aiming to reach low-income adults and children most at risk of diet-related disease, should explore the potential of these approaches to increase the accessibility and attractiveness of their programs. 

## Figures and Tables

**Figure 1 nutrients-12-02899-f001:**
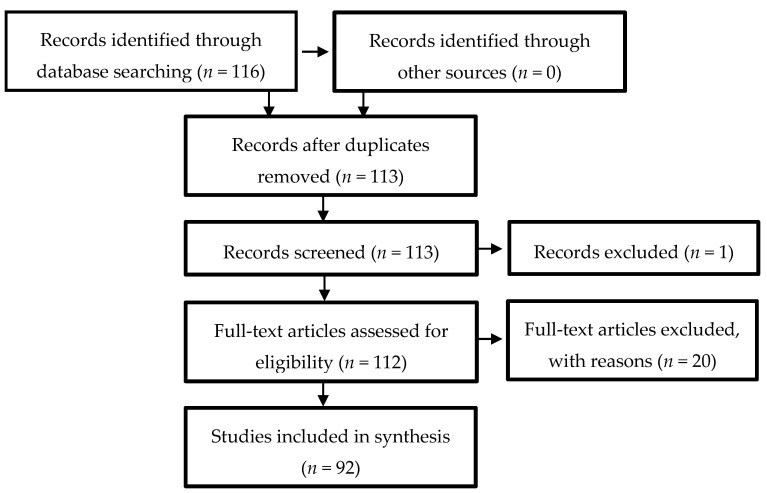
Preferred Reporting Items for Systematic Reviews and Meta-Analyses (PRISMA) flow diagram of literature search and selection process.

**Figure 2 nutrients-12-02899-f002:**
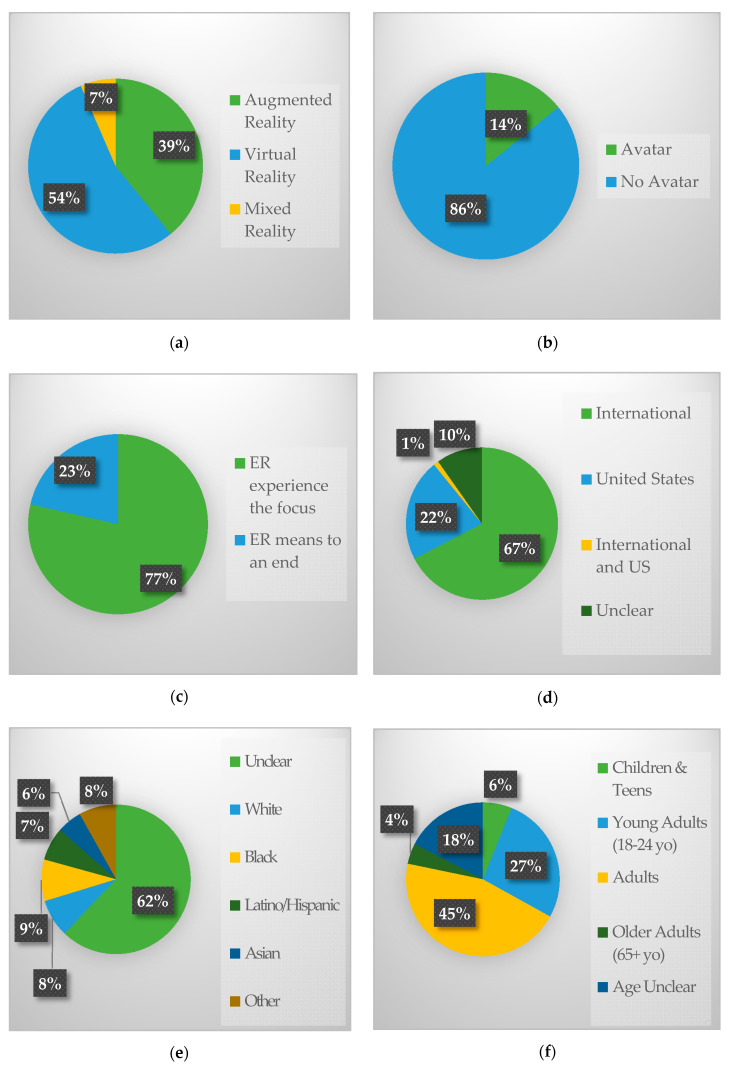
Summary of study characteristics by (**a**) technology type, (**b**) use of avatar, (**c**) purpose of technology, (**d**) location of study, (**e**) participant race, (**f**) participant age, (**g**) setting, (**h**) study design, (**i**) research focus), (**j**) outcome focus, (**k**) main equipment used.

**Figure 3 nutrients-12-02899-f003:**
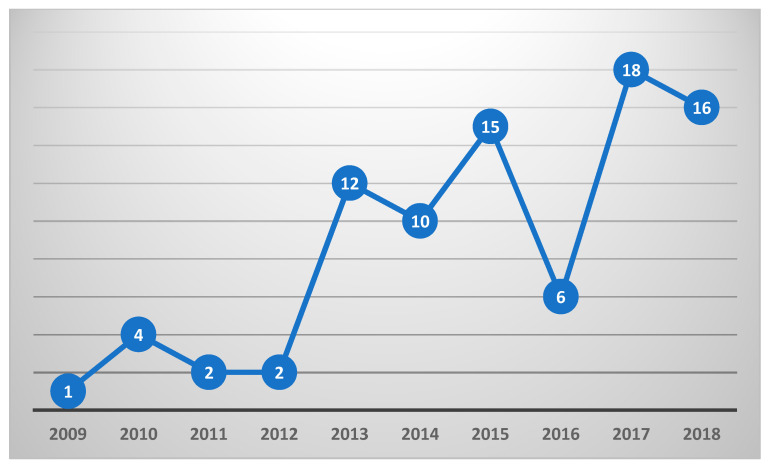
Publication trend from 2009–2018.

**Figure 4 nutrients-12-02899-f004:**
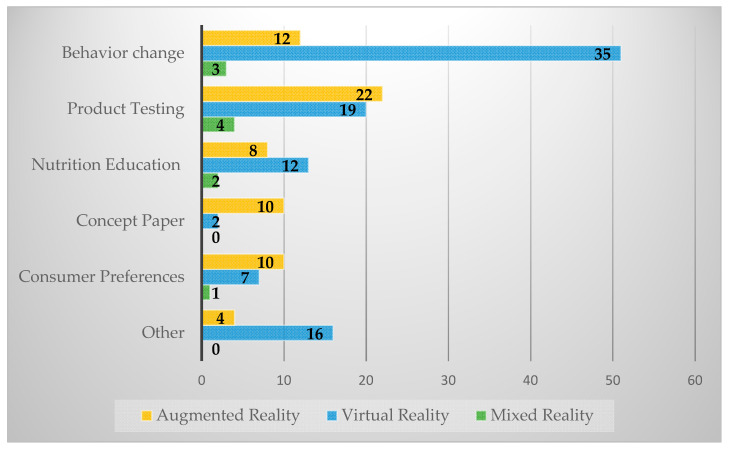
Study objectives by technology type (one type of tech in a study may contain multiple objectives).

**Figure 5 nutrients-12-02899-f005:**
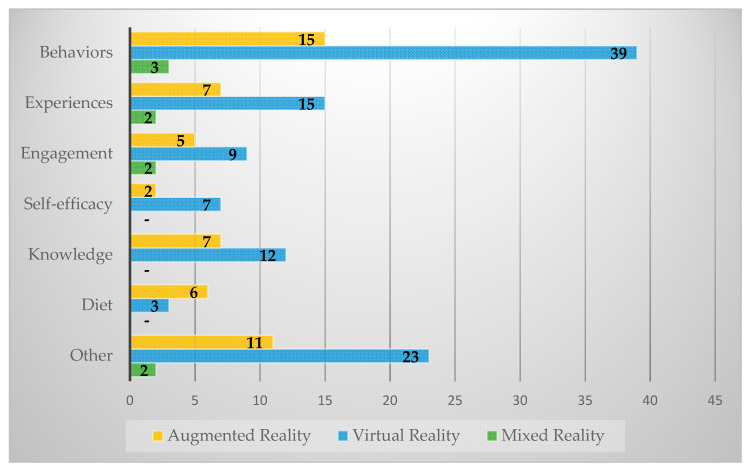
Technology by Study Outcomes.

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
