# Peer review of "Extended Reality Technologies in Nutrition Education and Behavior: Comprehensive Scoping Review and Future Directions"

_nutrients, 2020, doi:10.3390/nu12092899_

Round 1
Reviewer 1 Report
This manuscript presents findings from a scoping review of current published research regarding the use of extended reality technologies for education and behavior change. The Use of XR has enormous potential to aid development of new nutrition education and/or behavior change interventions. Consequently, a review of the literature is valuable. I enjoyed reading the paper and found it useful. I only have a few comments – generally just to expand on some aspects of the introduction and conclusion.
The introduction is clearly written and informative. There are some areas that may be interesting to cover in a little more detail. With respect to VR, a key advantage of this technology over traditional computer delivery is that it can create a sense of presence (the individual suspends belief and believes they are actually present in the VR scene). This may be key to the success of behavior change interventions and may be useful for education efforts. How many studies measured user prescence?
The authors state there is different equipment available at different price points. How does this influence the efficacy of the approach? Do participants using a mobile phone based headset feel the same level of presence and immersion as those using a higher end headset such as Vive or Rift? The authors do address this point but it may be useful to expand as many do not fully understand XR technology yet.
In addition, while headsets are generally used it is also possible to use CAVE systems – is there potentially a benefit to this in terms of presence and immersion? It may also be interesting to detail the different types of technology used (i.e., types of headset) in studies in the results section.
For the search terms used in the scoping review – I believe one or two studies have used supermarkets simulations in VR studies. While these studies may fall outside the scope of the review I suggest it might be another term to use in the search strategy.
The conclusion discusses many areas to move the research forward. Again, this is clearly written and raises several important issues. One area that would be potentially useful (in terms of future research) is a discussion of the practical use of this technology. For instance, how long can clients/patients use the headset without getting fatigued? If the counsellor/dietitian/nutritionist share the same VR space with their client how long can they be in the system until they become fatigued? Each of the XR modalities will likely be useful in different settings but, as for the introduction, within modalities there are significant differences in the specifications of equipment which may influence the effectiveness of the approach. I agree with the authors that these technologies may be potentially a significant advance in nutrition education but I also think the effect on the users (on all sides) needs further research.
Author Response
|
Reviewer 1 |
With respect to VR, a key advantage of this technology over traditional computer delivery is that it can create a sense of presence (the individual suspends belief and believes they are actually present in the VR scene). This may be key to the success of behavior change interventions and may be useful for education efforts. How many studies measured user presence? |
Thank you for your suggestion. We added the following to the paper:
“Perceived user presence in the virtual environment was mentioned by relatively few studies (n=13).” |
|
Reviewer 1 |
The authors state there is different equipment available at different price points. How does this influence the efficacy of the approach? Do participants using a mobile phone-based headset feel the same level of presence and immersion as those using a higher end headset such as Vive or Rift? The authors do address this point but it may be useful to expand as many do not fully understand XR technology yet. In addition, while headsets are generally used it is also possible to use CAVE systems – is there potentially a benefit to this in terms of presence and immersion? It may also be interesting to detail the different types of technology used (i.e., types of headset) in studies in the results section. |
This is a great point and we appreciate the suggestion. We have added the following to the manuscript: “VR can be accessed with a computer program (e.g. 3D models or avatars), simple VR viewers connected with mobile phones, or with a more dedicated head mounted display (HMD) or helmet mounted display (HeMD). AR can be accessed using mobile phone applications, AR smart eyeglasses, or dedicated headsets. Mixed reality is accessed using a dedicated MR headset. More easily accessed and affordable devices like computer experiences and mobile phone applications increase accessibility but may decrease immersion and user sense of presence compared to more expensive headset devices. Greater user presence may enhance applied effectiveness. A meta-analysis of the effect of immersive technology on user presence found that the technological immersion has a medium-sized effect on presence, and that individual immersive features such as user-tracking, stereoscopic visuals, and wider fields of view of visual display are particularly impactful features on the level of presence.”
We added the following to the Results section:
“The most common devices used for the XR experience included computers (n=34), mobile apps (n=25), head mounted devices (n=17), glasses (n=4), smart utensils (n=3), smart tables (n=2), gaming devices (n=2), and a few studies were unclear (n=7). Many studies (n=25) used multiple devices to create their user experience.”
|
|
Reviewer 1 |
For the search terms used in the scoping review – I believe one or two studies have used supermarkets simulations in VR studies. While these studies may fall outside the scope of the review I suggest it might be another term to use in the search strategy. |
Thanks for this suggestion. Our search terms were able to identify a few studies using supermarket simulations in VR studies, and those are represented in our findings. We have tried to clarify this on Lines 199 and 236 that the retail settings included places like virtual supermarkets. |
|
Reviewer 1 |
One area that would be potentially useful (in terms of future research) is a discussion of the practical use of this technology. For instance, how long can clients/patients use the headset without getting fatigued? If the counsellor/dietitian/nutritionist share the same VR space with their client how long can they be in the system until they become fatigued? Each of the XR modalities will likely be useful in different settings but, as for the introduction, within modalities there are significant differences in the specifications of equipment which may influence the effectiveness of the approach. I agree with the authors that these technologies may be potentially a significant advance in nutrition education but I also think the effect on the users (on all sides) needs further research. |
This is a great point.
We added the following to the Results section: “Only two studies reported information about fatigue, focusing on visual fatigue.”
We added the following to the Discussion section:
“The practical use of this technology for nutrition education and behavior is important to achieve wide-scale use in community settings. We found that very few articles reported factors like fatigue (n=2) from performing necessary repetitive tasks, headset discomfort, headache, nausea, and eye strain from screen viewing. Fatigue could influence the duration of use and enjoyment of using device-based technologies like XR.[22–24] These issues could vary based on XR modality and tasks required for the experience. Considerations of fatigue should be made for all end-users including consumers and education program participants of varying age groups, as well as counselors, educators, and clinicians. A recent study among a sample of children using AR found that fatigue was highest under the gesture condition, and lowest under the controller condition, and that this led to controller conditions being reported as the most usable by participants.[25] A study of VR devices found that despite advances in technology, head mounted display and helmet mounted display systems create visual discomfort after extended usage.[26] To determine if periods of rest could help visual fatigue during head mounted display VR experiences, Guo et al found that short rest during continuous use of the head mounted display induced more severe symptoms of subjective visual discomfort, but reduced objective visual fatigue.[27] Future research should continue to examine this issue of fatigue and ways that technology can reduce it. This will be particularly important for the wide-scale uptake of XR nutrition education and behavior approaches in community settings. ” |
Reviewer 2 Report
This was an interesting scoping review on the use of XR technology in nutrition education.
- As the authors bring up the potential for this technology to assist in bringing education to low-income communities due to issues with time, transportation etc in the introduction, it would be helpful to report on this outcome in the results section. The authors did comment that there was little published on sociodemographic characteristics in the discussion. However, I would recommend adding what was reported in the results and possibly a more robust discussion if there were any additional findings.
2. I would also recommend adding what the yes/no refers to within figure b since the note that explains it is so far below the figure. I would also recommend evenly spacing labels in figure d
Author Response
Thank you so much for taking the time to review our manuscript! Your time and feedback are greatly appreciated!
|
Reviewer 2 |
As the authors bring up the potential for this technology to assist in bringing education to low-income communities due to issues with time, transportation etc in the introduction, it would be helpful to report on this outcome in the results section. The authors did comment that there was little published on sociodemographic characteristics in the discussion. However, I would recommend adding what was reported in the results and possibly a more robust discussion if there were any additional findings. |
Thank you for this suggestion. We added the following to the results:
“Few studies reported income information (n=7), and only 3 studies reported focusing on low-income populations.”
We also highlighted this lack of focus on low-income populations in multiple parts of the Discussion. |
|
Reviewer 2 |
I would also recommend adding what the yes/no refers to within figure b since the note that explains it is so far below the figure. I would also recommend evenly spacing labels in figure d |
Thank you for this suggestion. We added more detail to the figure and improved the spacing as suggested. |
Reviewer 3 Report
The current work needs some extensive revisions before its publications. These are the followiing:
Specific research questions are missing in order to have a more concise presentation of results. This is the main reason of conducting a systematic review.
How students try to include only unbiased previous studies?
Are there any inclusion and exclusion criteria?
The authors refer that “This study has limitations. One common limitation of scoping reviews of the literature is publication bias, as the papers which we reviewed may have been published due to having certain outcomes”. So, we have read a systematic or a scoping review? Authors should clarify in specific their limitations. If such limits exist, what are the consequences in results?
Implications for practice and research (or design) about XR should be provided.
Author Response
Thank you so much for taking the time to review our manuscript! Your time and feedback are greatly appreciated!
|
Reviewer 3 |
How students try to include only unbiased previous studies? |
We would love to address this comment but feel unsure about how to interpret the comment given the wording, which we find unclear. If clarity can be provided that would be helpful. If the comment is concerning bias, scoping reviews typically do not contain a risk of bias assessment, which according to Viswanathan et al (2017), little conclusive empirical evidence exists on the validity of such assessments. To combat selection bias, we used a comprehensive and structured search and identified and included all papers that met our inclusion/exclusion criteria. We added the following to the paper:
“To reduce bias, we used a comprehensive and structured search and identified and included all papers that met our inclusion and exclusion criteria.” |
|
Reviewer 3 |
Are there any inclusion and exclusion criteria? |
Yes, we included inclusion and exclusion criteria. We had the following in the manuscript:
2.2. Search Strategy We conducted an extensive search of online journal and research databases, focusing on PubMed and Scopus results. Our search of the literature included two domains: XR approaches and dietary behaviors. We developed a search strategy with the guidance of an experienced academic research health sciences librarian, ultimately using Boolean operators and searched within subject heading, title, and abstract field. The search terms were developed after expert review and consensus and included: augmented reality or virtual reality or mixed reality or avatar AND diet* or nutrition or buffet or restaurant or grocery AND NOT farm. We searched in the English language, with a time filter of the past decade (1/1/2009 and later); articles had to be referenced in the research databases we selected and meet the inclusion criteria. To reduce potential selection bias, we applied no restrictions on article type or clinical population. 2.3. Selection of articles We included all research design types, including randomized controlled trials, quasi-experimental studies, observational studies, and product development/user testing. We focused on studies that used XR for nutrition education and nutrition behavior change as both the primary focus or the means to an end (the technology was used to achieve a condition with the experience or impact of XR not the focus). We excluded research that was a review of previous research that did not have research subjects and research that did not pertain to nutrition education or behaviors. |
|
Reviewer 3 |
The authors refer that “This study has limitations. One common limitation of scoping reviews of the literature is publication bias, as the papers which we reviewed may have been published due to having certain outcomes”. So, we have read a systematic or a scoping review? Authors should clarify in specific their limitations. If such limits exist, what are the consequences in results? |
We apologize for the confusion. We have tried to clarify that this is a scoping review rather than a systematic review by removing the use of systematic to avoid confusion. That term was used to describe the detailed and structured process we used. We also want to point to this part of the manuscript:
“For this paper, we used a scoping review process, which is a rigorous and transparent method that seeks to provide an overview of a potentially large and diverse body of literature pertaining to a broad topic, rather than a systematic review which attempts to collate empirical evidence from a relatively smaller number of studies pertaining to a focused research questions.[13]
We also added the following regarding the consequences of publication bias:
“One common limitation of scoping reviews of the literature is publication bias, as the papers which we reviewed may have been published due to having certain outcomes, which could lead to unbalanced reporting of findings and bias in favor of positive results.” |
|
Reviewer 3 |
Implications for practice and research (or design) about XR should be provided. |
Thank you for this suggestion. In addition to the topics included in the Discussion section, including the need for more rigorous and robust research methodology, and additional research among children and in community settings, we have added implications for future research regarding practical use and issues of fatigue. |
Reviewer 4 Report
I have not had time to read thoroughly, however I've identified a couple of issues. First of all I cannot find any description of what the actual scientific purpose of the survey was and the conclusions seem a bit odd, saying that since there is a lot of work done on XR for nutrition then it must be good to do XR for nutrition. This is a very unscientific way of thinking. It could be the other way around! Second, I cannot find any list of the actual papers surveyed and thereby also no table of how they were classified. The third main issue I identified was that half of the references used as scientific basis for the discussion are not published in peer review journal or conference proceedings.
Author Response
Thank you so much for reviewing our paper! We greatly appreciate the time and feedback!
|
Reviewer 4 |
First of all I cannot find any description of what the actual scientific purpose of the survey was |
Thank you for raising this issue. We would love to clarify this for you. On lines 75-85, we have the following:
“At this point, it is unclear how these technologies have been used for nutrition education and behavior change and what gaps remain in the development and implementation of XR programming. Thus, there is a need for an examination of the literature around the use of XR technologies in nutrition education and behavior change to identify gaps and needs for future work. A previous paper,[11] which summarized background information and discussions for a joint July 2010 National Institutes of Health — US Department of Defense workshop, discussed research needs and opportunities for VR technologies for research and education in obesity and diabetes, but the report focused solely on VR for obesity and diabetes. Since the publication of that report, various iterations of XR, including AR and MR, have emerged, and the potential of this technology is more fully realized.” |
|
Reviewer 4 |
The conclusions seem a bit odd, saying that since there is a lot of work done on XR for nutrition then it must be good to do XR for nutrition. This is a very unscientific way of thinking. It could be the other way around! |
Thank you for your response. We modified our wording to try and more clearly express our position:
“Our findings suggest that the integration of XR technology in the field of nutrition education and behavior change is a burgeoning approach that may have the potential to be successfully used to influence positively impact diet-related behaviors, though the efficacy and effectiveness of the impact on diet related behaviors remains unclear.” |
|
Reviewer 4 |
Second, I cannot find any list of the actual papers surveyed and thereby also no table of how they were classified. |
The actual papers surveyed were provided in a Supplementary table. We have added the following statement to clarify:
“2.6. Data Availability. For a listing of reviewed manuscripts and data not shown in the Results section, please see the Supplementary Materials.”
|
|
Reviewer 4 |
The third main issue I identified was that half of the references used as scientific basis for the discussion are not published in peer review journal or conference proceedings |
Thank you for this feedback. We have added additional references to support the Discussion as best as possible. Given that this is a relatively new topic in the literature, finding references is challenging, but we tried the best that we could to select references that fit the topics being discussed.
|
Round 2
Reviewer 3 Report
The authors have made a considerable work to revise their manuscript. Therefore, I would like to recommends its acceptance.
Author Response
Thank you! We greatly appreciate your time and effort in reviewing our paper!